# The effect of explicit convection on simulated malaria transmission across Africa

**Joshua Talib** [1]*, **Abayomi A. Abatan**[2], **Remy HoekSpaans**[3], **Edmund I. Yamba**[4], **Temitope S. Egbebiyi**[5], **Cyril Caminade**[6,7], **Anne Jones**[8], **Cathryn E. Birch**[9], **Oladapo M. Olagbegi**[10], **Andrew P. Morse**[11]

**1** U.K. Centre for Ecology and Hydrology (UKCEH), Wallingford, United Kingdom, **2** College of Engineering, Mathematics and Physical Sciences, University of Exeter, Exeter, United Kingdom, **3** Liverpool School of Tropical Medicine (LSTM), Liverpool, United Kingdom, **4** Department of Meteorology and Climate Science, Kwame Nkrumah University of Science and Technology (KNUST), Kumasi, Ghana, **5** Climate Systems Analysis Group, Department of Environmental and Geographical Science, University of Cape Town, Cape Town, South Africa, **6** Institute of Infection, Veterinary and Ecological Sciences, University of Liverpool, Liverpool, United Kingdom, **7** The Abdus Salam International Centre for Theoretical Physics (ICTP), Trieste, Italy, **8** International Business Machines (IBM) Research Europe, Daresbury, United Kingdom, **9** School of Earth and Environment, University of Leeds, Leeds, United Kingdom, **10** School of Health Sciences, University of KwaZulu-Natal, Durban, South Africa, **11** School of Environmental Sciences, University of Liverpool, Liverpool, United Kingdom

* jostal@ceh.ac.uk

**Data Availability Statement:** All data used in this study is freely available. Malaria simulation outputs underlying the presented results are available at https://doi.org/10.6084/m9.figshare.24565135.v2.

## Abstract

Malaria transmission across sub-Saharan Africa is sensitive to rainfall and temperature. Whilst different malaria modelling techniques and climate simulations have been used to predict malaria transmission risk, most of these studies use coarse-resolution climate models. In these models convection, atmospheric vertical motion driven by instability gradients and responsible for heavy rainfall, is parameterised. Over the past decade enhanced computational capabilities have enabled the simulation of high-resolution continental-scale climates with an explicit representation of convection. In this study we use two malaria models, the Liverpool Malaria Model (LMM) and Vector-Borne Disease Community Model of the International Centre for Theoretical Physics (VECTRI), to investigate the effect of explicitly representing convection on simulated malaria transmission. The concluded impact of explicitly representing convection on simulated malaria transmission depends on the chosen malaria model and local climatic conditions. For instance, in the East African highlands, cooler temperatures when explicitly representing convection decreases LMM-predicted malaria transmission risk by approximately 55%, but has a negligible effect in VECTRI simulations. Even though explicitly representing convection improves rainfall characteristics, concluding that explicit convection improves simulated malaria transmission depends on the chosen metric and malaria model. For example, whilst we conclude improvements of 45% and 23% in root mean squared differences of the annual-mean reproduction number and entomological inoculation rate for VECTRI and the LMM respectively, bias-correcting mean climate conditions minimises these improvements. The projected impact of anthropogenic climate change on malaria incidence is also sensitive to the chosen malaria model and representation of convection. The LMM is relatively insensitive to future changes in precipitation intensity, whilst VECTRI predicts increased risk across the Sahel due to enhanced

Observations including CHIRPS precipitation and ERA5 temperatures can be downloaded at https://www.chc.ucsb.edu/data/chirps and https://cds.climate.copernicus.eu/cdsapp\#!/home. Additional observational data used for analysis in the supplementary information including BEST and ISIMIP2b atmospheric data can be found at https://berkeleyearth.org/data/ and https://dataservices.gfz-potsdam.de/pik/showshort.php?id=escidoc:3928916 respectively. MetUM climate simulation data is publicly available at https://catalogue.ceda.ac.uk/uuid/a6114f2319b34a58964dfa5305652fc6. MAP data is freely accessed at https://data.malariaatlas.org/maps. The two malaria models utilised in this study are publicly available. VECTRI source code and documentation can be found at http://users.ictp.it/~tompkins/vectri/, whilst the LMM is available from https://doi.org/10.5281/zenodo.5494445.

**Funding:** "The British Council Researcher Links Climate Challenge Grant (715056901) provided funding for a workshop titled "Investigating the effects of climate change on malaria for urgent action to combat climate change with reference to COP26 priorities" and a follow-on project. Both supported this paper. For this study JT was supported by the Natural Environment Research Council (NERC) as part of the National Capability (NC) international programme (NE/X006247/1). AJ was supported by the Hartree National Centre for Digital Innovation, a collaboration between Science and Technology Facilities Council (STFC) and International Business Machines (IBM). The funders had no role in study design, data collection and analysis, decision to publish, or preparation of the manuscript."

**Competing interests:** The authors have declared that no competing interests exist.

rainfall. We postulate that VECTRI's enhanced sensitivity to precipitation changes compared to the LMM is due to the inclusion of surface hydrology. Future research should continue assessing the effect of high-resolution climate modelling in impact-based forecasting.

# 1 Introduction

Malaria is an infectious disease which is transmitted by female mosquitoes of the *Anopheles* species. Whilst initial symptoms are mild, if not treated malaria can lead to severe illness and death [1]. In 2020 there were approximately 241 million malaria cases worldwide, with approximately 95% of these cases occurring in Africa [2]. Even with substantial efforts to alleviate the societal burden of malaria [3–6], the disease is still responsible for more than half a million deaths across Africa every year, with approximately 80% of fatalities being children aged under 5 [2].

As climate conditions influence multiple components of an Anopheles mosquito's life cycle, malaria transmission is highly sensitive to local weather conditions [7–9]. Firstly, the laying of eggs at the beginning of a mosquito's life cycle relies on open water bodies and substantial precipitation accumulations [10, 11]. Under a dry environment a relatively small number of eggs are laid, whilst excessive rainfall can wash away breeding sites altogether [12]. Mosquito survival during other aquatic life cycle components, including the egg, larval and pupal stage, also depends on precipitation totals [13–15]. For example, minimal precipitation after the laying of eggs can lead to overcrowding and poor water quality. Adult mosquitoes are less sensitive to rainfall and more dependent on temperature and relative humidity [8, 16, 17]. Daily-mean temperatures outside of 5 to 40°C are fatal for adult *Anopheles* mosquitoes [14], whilst the largest probabilities for *Anopheles* survival has been observed and modelled between 10 and 25°C [8, 16, 18]. As well as climate conditions influencing stages of a mosquito's life cycle, the development of the malaria parasite within the mosquito itself, often referred to as the sporogonic cycle, is also sensitive to temperature [16, 17, 19]. The majority of studies conclude that the development of most *Plasmodium* malaria parasites ceases below temperatures between 14.5 and 19°C [20–22]. The multiple mechanisms with which weather conditions affect malaria transmission leads to a non-linear relationship [9], making it non-trivial to predict the influence of weather conditions on malaria incidence. As well as climate-driven uncertainties, non-climatic environmental factors such as water quality, food supply, predators and variability amongst mosquito species, increase the complexity at predicting malaria transmission [23–26]. Additionally, social-economic factors impacting the administration of disease control measures, including insecticide spraying, bed nets and treatment of cases, can vary the severity and number of malaria infections [27–29].

Despite the numerous complexities involved in predicting malaria incidence, several research groups have developed models for malaria transmission [8, 14, 28, 30, 31]. These models have been used to predict malaria transmission at various timescales including sub-seasonal and decadal [32–34]. Global and regional climate change risk assessments investigating the impact of a warmer climate on malaria transmission predict decreased prevalence across the majority of Africa [35–38] due to higher temperatures restricting host seeking [39], blood feeding [19] and mosquito development [14, 16]. Increased temperatures only increase malaria transmission risk across highland regions of eastern and southern Africa [33, 37, 40, 41]. However, differences between malaria models, greenhouse gas emission scenarios, and projected circulation changes lead to large uncertainties in future malaria risk estimates [33]. Whilst the largest uncertainties are associated with the different malaria impact models,

differences between future atmospheric conditions projected by general circulation models (GCMs) lead to significant uncertainties across regions with substantial changes in malaria transmission, such as the East African (EA) highlands and northern Sahel [33]. Constraining uncertainty originating from GCM outputs will support efforts to predict malaria transmission in a warmer climate.

Currently, predictions of malaria transmission under present and future climates have only been performed using relatively coarse-resolution regional or global climate models [30, 33, 35, 36, 42]. Such models require a parameterisation of convection, atmospheric vertical motion driven by spatial temperature variations and responsible for intense rainfall, and are known to poorly represent tropical precipitation characteristics including rainfall frequency and extreme rainfall events [43, 44]. Finer-resolution convection-permitting simulations on the other hand, better represent intense rainfall and dry spells [45–47], alongside related processes including storm life cycle [48], the atmospheric water cycle [47, 49], and soil moisture-precipitation feedbacks [50, 51]. Additionally, convection-permitting simulations project larger increases in extreme rainfall [52–55], soil erosion [56], and humid heatwaves [57] across Africa under a warmer climate. However, whilst previous studies, including several that evaluated model simulations used in this study, show that explicitly representing convection better resolves rainfall frequency and intensity, such simulations have larger errors in 10-day precipitation accumulations and near-surface temperatures [43, 46, 47]. Therefore, as malaria transmission has a non-linear relationship with local weather conditions [9, 10, 21], it is unknown to what extent using climate data from explicit convection simulations changes predictions of malaria transmission. In this study we use two climate-sensitive dynamical malaria models, the Liverpool Malaria Model (LMM) and Vector-Borne Disease Community Model of the International Centre for Theoretical Physics (VECTRI), to investigate the effect of explicitly representing convection on simulated malaria transmission. We also compare malaria model outputs driven with observations and climate model data to consider the added-value of high-resolution convection-permitting simulations at supporting malaria predictions.

## 2 Methodology

### 2.1 Malaria transmission models

Two climate-sensitive dynamical malaria models, VECTRI [31] and LMM [15, 32], are used to investigate the effect of explicitly representing convection on simulated malaria transmission. Both models are driven by the same climate variables: daily-accumulated precipitation (mm day$^{-1}$); and daily-mean near-surface (2 m) air temperature (˚C). Whilst the LMM can only be used to investigate the climate suitability for malaria transmission, VECTRI also considers the impact of surface hydrology and population density.

We use a simplified version of the LMM [15, 32], which is a compartmental model of malaria transmission that computes the basic reproduction ratio, $R_0$. In epidemiology, $R_0$ quantifies the potential of a disease to spread in a fully susceptible population [58]. When $R_0$ is greater than 1, the size of the infected population is predicted to grow and the disease epidemic will persist. A full description of the simplified LMM can be found in supplementary section S1.1 in S1 File.

VECTRI is an open-source weather-driven malaria model developed to better understand the drivers of malaria transmission across Africa [31]. It is commonly used to produce national-level malaria predictions across Africa [59–61]. VECTRI's framework connects a biological model of mosquito and parasite life cycles to a secondary model which simulates disease prevalence amongst humans. A description of VECTRI can be found in supplementary section S1.2 in S1 File, whilst specific details on model formulation and parameter settings can be

found in [31]. VECTRI outputs several different parameters including the human biting rate and mosquito density. In this study we focus on analysing outputs of the entomological inoculation rate (EIR; infectious bites person$^{-1}$ day$^{-1}$), which is defined as the product of the human biting rate and sporozoite infection rate.

## 2.2 Observed malaria transmission

To validate our malaria simulations, we use estimates of newly diagnosed *Plasmodium falciparum* (Pf) cases from the Malaria Atlas Project (MAP) [62]. MAP is based on a Bayesian statistical model that estimates malaria incidence by amalgamating multiple data sources including: malaria control interventions such as insecticide-treated bed nets and indoor residual spraying; parasite incidence rates from over 27,000 population clusters and; environmental and socio-demographic covariates [29, 63, 64]. Focusing on environmental predictors, MAP utilises precipitation from WorldClim, which interpolates thousands of station observations [65], as well as surface temperature, wetness and land cover from Terra and Aqua Moderate Resolution Imaging Spectroradiometer (MODIS) satellite products [66]. Given that MAP output is only derived using observations of environmental conditions rather than climate model data, we cannot postulate that malaria model output driven by a certain climate model configuration will have a stronger agreement with MAP data. As employed malaria model outputs are different from each malaria model ($R_0$ and EIR; section 2.1), we compare model diagnostics to the MAP Pf rate in section 3.1.

## 2.3 Climate data

**2.3.1 Observational and reanalysis data.** In this study observations and reanalysis data are used to compare simulated malaria transmission when driving malaria models with either observations or climate model data. After comparing LMM experiments driven by different observed rainfall and temperature datasets with MAP malaria incidence data, we concluded that precipitation from the Climate Hazards group InfraRed Precipitation with Stations (CHIRPS) dataset and near-surface temperatures from the European Centre for Medium-Range Weather Forecasts (ECWMF) Reanalysis version 5 (ERA5) were the best (section S2 in S1 File). We also use these two products to evaluate simulated atmospheric conditions in climate model experiments (section S3 in S1 File).

CHIRPS is avaliable on a 0.05˚ latitude/longitude grid and is derived using a combination of satellite-derived infrared measurements and gauge-based rainfall totals [67]. Several studies have illustrated that CHIRPS is one of the most reliable pan-African precipitation products available [67–69]. Daily-means of near-surface temperatures on a 0.25˚ latitude/longitude grid were computed using hourly ERA5 data. ERA5, which is computed using four-dimensional variational data assimilation (4D-VAR) and cycle 41r2 of the Integrated Forecasting System (IFS), provides a detailed continuous record of the global atmosphere, land and ocean waves [70, 71]. Whilst the computation of ERA5 relies on parameterising environmental processes including atmospheric convection, previous studies have shown small differences between annual-mean temperatures from ERA5 and other gridded observational products [57, 72]. For the rest of this study we use the term "observations" to refer to both CHIRPS rainfall and ERA5 temperature datasets.

As well as using observed atmospheric data, the VECTRI malaria model requires human population density data. For both historical and future VECTRI experiments, we use estimated 2020 population counts from the Gridded Population of the World (GPW) version 4 dataset [73]. GPW is produced by the Socioeconomic Data and Applications Center (SEDAC) at the National Aeronautics and Space Administration (NASA), and utilises spatially-explicit

administrative boundary data and tabular population counts by local administrators [74]. All the collated data is regridded onto an approximately 1 km latitude/longitude grid using an areal-weighting method [75]. To be consistent with the resolution of climate simulation data (section 2.3.2), all datasets have been remapped onto the same 0.25˚ horizontal grid using a first-order conservative interpolation scheme [76].

**2.3.2 Climate model data.** In addition to driving malaria models with observations (section 2.3.1), we perform a set of malaria transmission experiments using climate model outputs from Met Office Unified Model (MetUM) pan-African (-45–40˚N, -25–57˚E) simulations. The MetUM is a non-hydrostatic model with a semi-implicit, semi-Lagrangian dynamical core. Full details of the model specifications and setup are provided by [54, 77].

We use four 10-year atmosphere-only UKMO simulations which either have a convection-permitting model set-up with a horizontal resolution of approximately 4.5 km (CP4), or a parameterised convection configuration with a grid spacing of approximately 25 km (R25). For each configuration, a historical (subscript h) and future (subscript f) climate simulation has been performed (Table 2). In this study R25 is treated as the "baseline" climate model, whilst CP4 is used to investigate the impact of explicitly representing convection on malaria transmission. For historical climate simulations, observed sea surface temperatures from [78] are prescribed. Meanwhile for future climate, prescribed SSTs are the sum of analyses from [78] over 1997–2007 and the climatological average SST change from 1975–2005 to 2085–2115 simulated by a Hadley Centre Global Environment Model version 2 Earth System model (HadGEM2-ES) under a representative concentration pathway (RCP) of 8.5 (RCP8.5) [79]. To ensure SST variability remains similar between the historical and future climate, monthly SST changes were interpolated spatially and temporally before being added to the daily-varying SST forcing data [78]. The difference in global-mean SST between the historical and future climate is just under 4˚C, whilst global mean air temperatures increase by 5.2˚C [80]. Focusing on prescribed atmospheric composition, in historical simulations $CO_2$ concentrations vary annual and increase by approximately 17 ppm between 1997 and 2007. Meanwhile for future climate simulations, GHG concentrations were taken from the RCP8.5 scenario for 2100. Table 1 summarises the model set-up for each UKMO simulation. To enable a fair comparison

**Table 1. Details of UKMO simulations whose outputs are used to drive the LMM and VECTRI.**

| Model configuration | CP4 | CP4 | R25 | R25 |
|---|---|---|---|---|
| Model experiment | $CP4_h$ | $CP4_f$ | $R25_h$ | $R25_f$ |
| Simulation length (years) | 10 | 10 | 10 | 10 |
| Simulated time period | 1997–2006 | 2097–2106 | 1997–2006 | 2097–2106 |
| Simulation domain | Pan-African | Pan-African | Pan-African | Pan-African |
| Nesting model | N512 MetUM GA7 simulation | N512 MetUM GA7 simulation | N512 MetUM GA7 simulation | N512 MetUM GA7 simulation |
| Horizontal resolution (km) | 4.5 | 4.5 | 26.0 | 26.0 |
| Time step (s) | 100 | 100 | 600 | 600 |
| Representation of convection | Explicit | Explicit | Parameterized based on [81] | Parameterized based on [81] |
| Number of vertical levels | 80 | 80 | 63 | 63 |
| Prescribed surface type | Sandy soils | Sandy soils | Sandy soils | Sandy soils |
| Prescribed SSTs | High-resolution analyses [78] | High-resolution analyses plus SST change in RCP8.5 HadGEM2-ES | High-resolution analyses [78] | High-resolution analyses plus SST change in RCP8.5 HadGEM2-ES |
| Prescribed greenhouse gas concentrations | Annual observed greenhouse gas concentrations | RCP8.5 scenario for 2100 [79] | Annual observed greenhouse gas concentrations | RCP8.5 scenario for 2100 [79] |

**Table 2. Malaria transmission experiments.**

| Purpose | Simulation name | Precipitation input data | Temperature input data |
|---|---|---|---|
| Malaria experiment using observations (LMM and VECTRI) | $O_h$ | CHIRPS | ERA5 |
| Sensitivity to representation of convection (LMM and VECTRI) | $C_h$ | $CP4_h$ | $CP4_h$ |
| | $R_h$ | $R25_h$ | $R25_h$ |
| | $C_f$ | $CP4_f$ | $CP4_f$ |
| | $R_f$ | $R25_f$ | $CP4_f$ |
| Isolating the importance of simulated temperature and precipitation differences (LMM only) | $CP_hRT_h$ | $CP4_h$ | $R25_h$ |
| | $RP_hCT_h$ | $R25_h$ | $CP4_h$ |
| | $CP_fCT_h$ | $CP4_f$ | $CP4_h$ |
| | $RP_fRT_h$ | $R25_f$ | $R25_h$ |
| | $CP_hCT_f$ | $CP4_h$ | $CP4_f$ |
| | $RP_hRT_f$ | $R25_h$ | $R25_f$ |
| Removing the influence of mean atmospheric model biases (LMM only) | $C_hBC$ | $CP4_hBC$ | $CP4_hBC$ |
| | $R_hBC$ | $R25_hBC$ | $R25_hBC$ |

between malaria simulations driven with CP4 and R25 data, CP4 data is remapped onto the same horizontal grid as R25 using a first-order conservative interpolation scheme [76]. In supplementary section S3 in S1 File, we provide a brief assessment of temperature and precipitation biases in historical MetUM simulations. In summary, and in agreement with previous studies [43, 46, 47], whilst explicitly representing convection improves the simulated frequency and intensity of daily precipitation accumulations, errors in near-surface temperatures and 10-day precipitation accumulations increase.

## 2.4 Malaria transmissions experiments

**2.4.1 Investigating the sensitivity of simulated malaria transmission to the representation of convection.** To compare differences in simulated malaria transmission when using observations or climate model data, we perform three historical simulations for each malaria model (Table 2). These simulations are either driven by observations ($O_h$), CP4 ($C_h$) or R25 ($R_h$) historical data, and enable us to understand the sensitivity of simulated malaria transmission to observed and model input data. For the rest of this study, "O", "C" and "R" are shorthand for observational, CP4 and R25 data respectively. For both VECTRI and the LMM, we also perform a set of simulations using future climate model data ($C_f$ and $R_f$). The use of future climate model simulations enables us to investigate the effect of explicitly representing convection in both present and future climates. In particular, it enables us to assess whether explicitly representing convection changes predictions of malaria transmission.

**2.4.2 Determining whether temperature or precipitation is responsible for simulated malaria transmission differences.** To isolate the importance of differences in simulated temperature and precipitation when changing the representation of convection, we perform simulations with either temperature or precipitation sourced from $CP4_h$, with the remaining variable sourced from $R25_h$ (Table 2). We also perform sensitivity experiments with which we only use precipitation or temperature from the future climate. Whilst we note that precipitation and temperature are not independent of each other, sensitivity experiments enable an initial understanding of the key drivers responsible for differences in malaria predictions. The labelling of all sensitivity experiments follows the same structure and includes the source of precipitation (P) data followed by the driving temperature (T) data. For example, $RP_fRT_h$ denotes that precipitation and temperature data is sourced from future and historical R25

experiments respectively. Due to limited computational resources, sensitivity experiments are only performed using the LMM.

**2.4.3 Understanding the importance of temperature and precipitation mean biases in malaria transmission simulations.** Given substantial errors in simulated precipitation and temperature in both climate model configurations (section S3 in S1 File), we also perform LMM experiments driven with bias-corrected, linearly-scaled simulation data. For temperature and precipitation data, we applied an additive and relative adjustment respectively [82–84]. The use of linear-scaling only corrects time-mean biases, hence, variability errors still persist. LMM experiments driven with bias-corrected $CP4_h$ and $R25_h$ data are labelled as $C_hBC$ and $R_hBC$ respectively. All malaria simulations are summarised in Table 2.

## 3 Results

### 3.1 Evaluation of present-day malaria predictions

Before investigating the effect of explicit convection on predicted malaria transmission, we first assess whether our malaria models (section 2.1) simulate realistic patterns of malaria incidence by comparing historical malaria model simulations ($O_h$, $C_h$ and $R_h$) with estimates of malaria endemicity from MAP [62]. Fig 1a shows the estimated annual-mean incidence rate of Pf, the most common malaria parasite to infect humans across Africa [85], between years 2000 to 2007. Unfortunately, we are unable to validate the use of VECTRI and LMM using the same variables due to different model hypotheses, and consequently, different model diagnostics.

Even though the LMM is solely driven by climate variables and does not take into account social-economic factors (section 2.1 and section S1 in S1 File), there is good agreement between LMM-simulated malaria transmission and the MAP-estimated Pf incidence rate ($r = 0.45$, Fig 1a and 1b). High Pf incidence and LMM-simulated $R_0$ is seen across central parts of Africa, the Guinea coast, and south-eastern countries such as Malawi and Mozambique. However, substantial discrepancies in predicted malaria transmission occurs across the Guinea coast. EIR model output from VECTRI on the other hand, has a better agreement with large MAP-estimated Pf incidence rates across West Africa and the Guinea coast (Fig 1e). This suggests that low malaria incidence across the Guinea coast in LMM experiments is associated with a lack of socio-economic factors in the design of the LMM. Outputs from the LMM and VECTRI driven with CP4 or R25 data (Fig 1c, 1d, 1f and 1g) also show a realistic distribution of malaria transmission risk with respect to the MAP-estimated Pf incidence rate. All malaria model experiments, regardless of the chosen driving data or malaria model, have significant ($p \leq 0.01$) spatial correlations with Pf incidence data with correlation coefficients ranging from 0.37 to 0.45. Given the good agreement between malaria model outputs and MAP-estimated incidence rates, the effect of using explicit convection climate model data is investigated further using both malaria models in the following subsection.

### 3.2 The sensitivity of simulated malaria transmission to the representation of convection

To understand whether differences in the representation of convection affects the simulation of malaria transmission, we analyse simulated transmission differences when driving the LMM and VECTRI with observational, $CP4_h$ or $R25_h$ data. Comparing $C_h$ and $R_h$ LMM experiments highlights a west-to-east dipole difference in predicted malaria risk (Fig 2a). Across lowland regions in the west, including parts of the Guinea coast and Congolese rainforests, malaria risk is higher in $C_h$, meanwhile across eastern highland regions, simulated malaria transmission risk is smaller. Focusing on LMM differences between $C_h$ and $R_h$ compared to

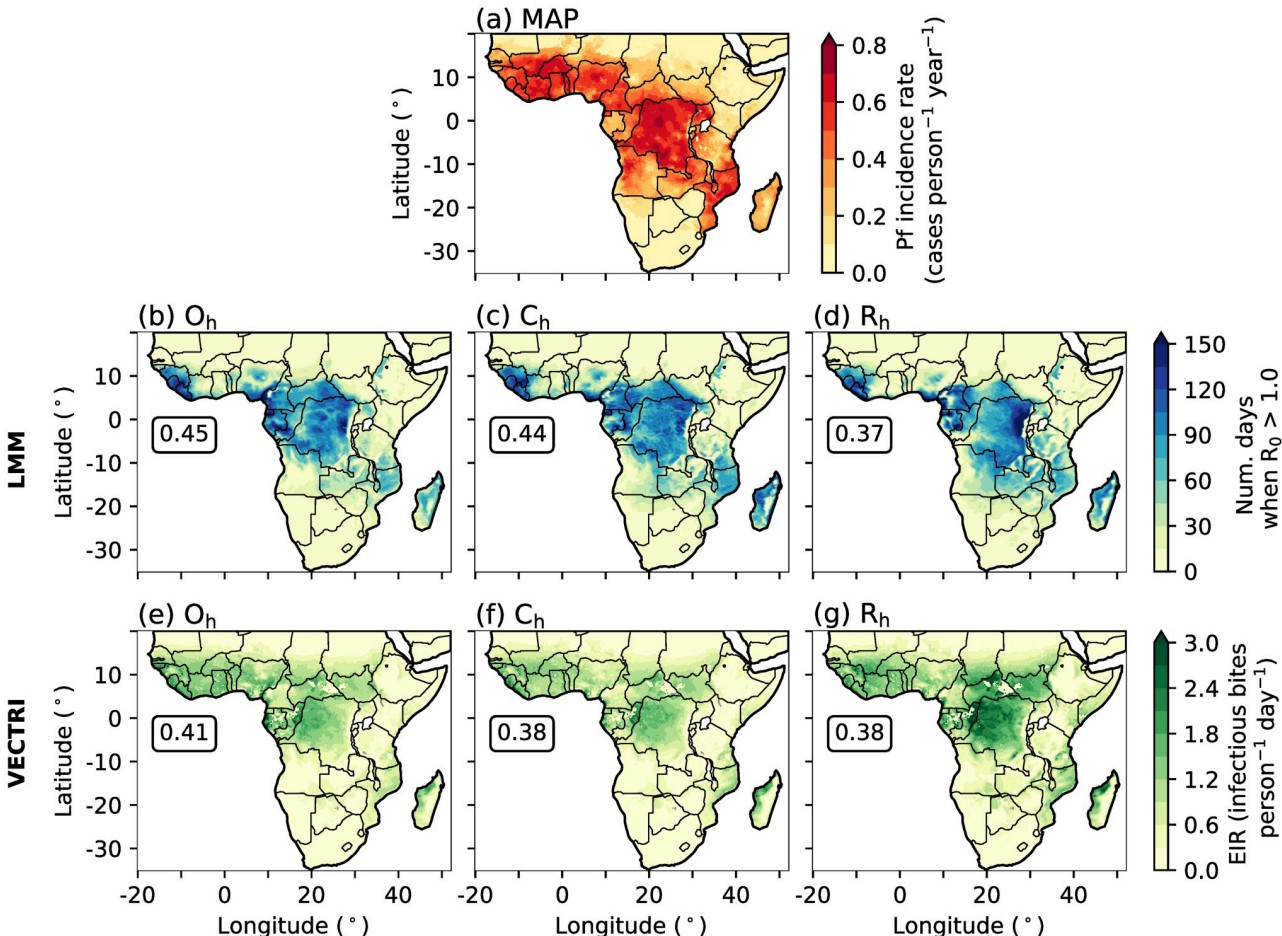

**Fig 1.** Annual-mean (2000–2007) of (a) Pf incidence rate (cases person$^{-1}$ year$^{-1}$) from MAP, (b-d) number of days when $R_0$ is greater than 1.0 from LMM experiments, and (e-g) predicted EIR (infectious bites person$^{-1}$ day$^{-1}$) from VECTRI simulations. We show outputs from LMM and VECTRI driven by (b,e) observational, (c,f) CP4$_h$ and (d,g) R25$_h$ data. In panels (b) to (g) we document the spatial correlation coefficient between simulated malaria model outputs and MAP data. To ensure that spatial correlations are not biased towards regions of low malaria incidence, we remove all grid points where the MAP-derived Pf incidence rate is smaller than 0.1. We also removed grid points where the simulated annual-mean number of days when $R_0$ is greater than 1.0 is outside the range of 15.0 and 140.0, or where the simulated EIR is outside 0.3 and 2.0. To be consistent with the time span of MAP data [62], we only analyse malaria model output which is driven with observations or climate model data from years 2000 to 2007. All spatial correlations are statistically significant at a 99% confidence interval using a two-tailed Wald T-test. Land and country boundaries were added using Natural Earth; free vector and raster map data available at naturalearthdata.com.

$O_h$ (Fig 2b and 2c), it is evident that $C_h$ has a stronger agreement with $O_h$ than $R_h$. The RMSD on the annual-mean number of days when $R_0$ is greater than 1.0 is approximately 23% smaller in $C_h$ than in $R_h$. $R_h$ overestimates the number of days when malaria transmission is increasing across highland regions of East Africa, Angola and Zambia. It also underestimates malaria risk across parts of central Africa and the Guinea coast. Differences between $C_h$ and $R_h$ LMM outputs highlight that the representation of convection can play a substantial role in the magnitude of simulated malaria transmission.

Sensitivity experiments which use either temperature or precipitation data from CP4$_h$ highlight that temperature differences between the two MetUM configurations are mostly responsible for differences in simulated malaria transmission by the LMM (Fig 2d and 2e). The difference in simulated malaria risk when using CP4$_h$ temperatures is similar to the difference when changing both atmospheric variables (Fig 2a and 2d). However, differences between

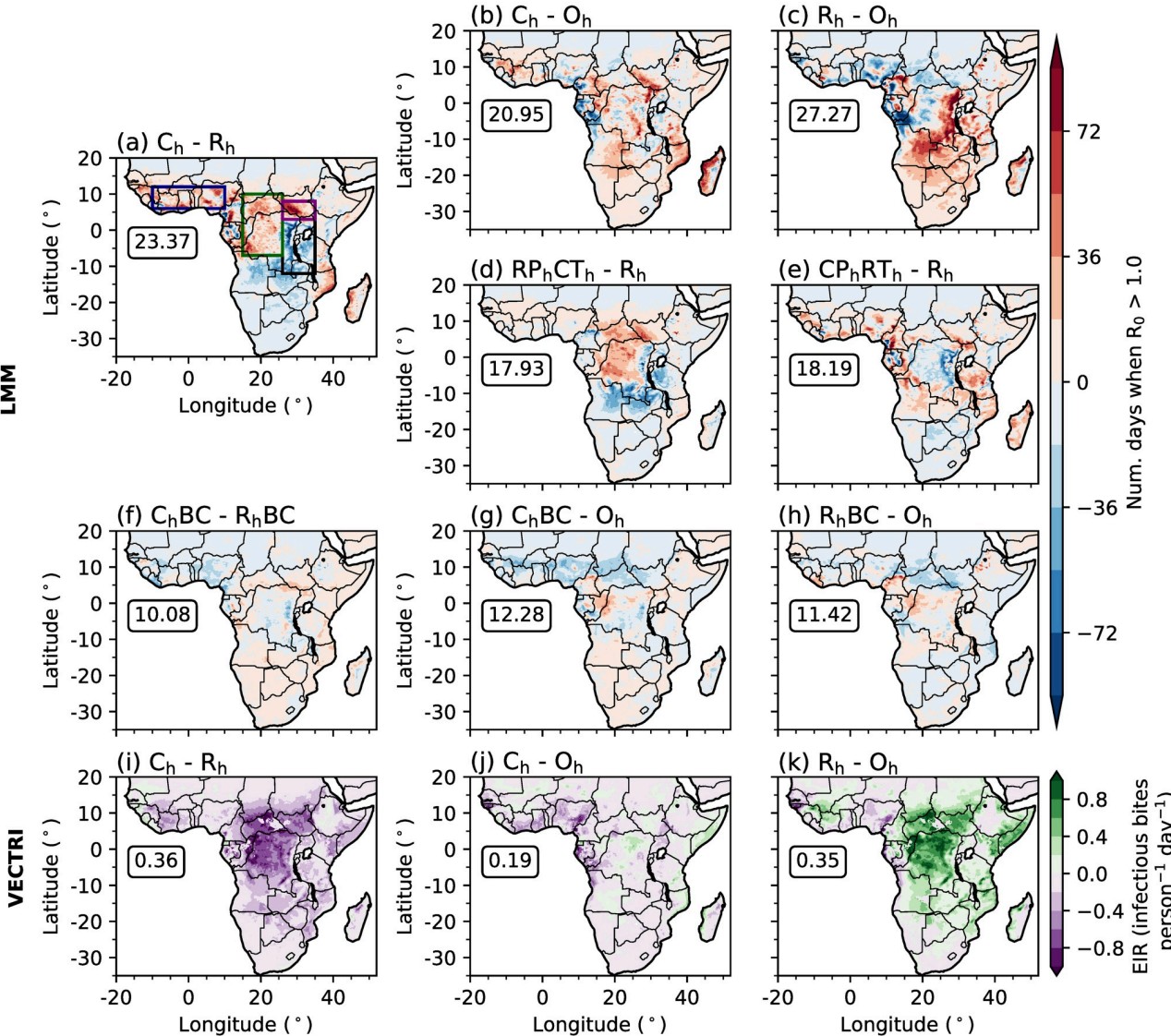

**Fig 2.** Differences in the simulated annual-mean number of days when $R_0$ is greater than 1.0 between (a) $C_h$ and $R_h$, (b) $C_h$ and $O_h$, (c) $R_h$ and $O_h$, (d) $RP_hCT_h$ and $R_h$, (e) $CP_hRT_h$ and $R_h$, (f) $C_hBC$ and $R_hBC$, (g) $C_hBC$ and $O_h$, and (h) $R_hBC$ and $O_h$, by the LMM. (i-k) Differences in the simulated annual-mean EIR (infectious bites person$^{-1}$ day$^{-1}$) between (i) $C_h$ and $R_h$, (j) $C_h$ and $O_h$, (k) $R_h$ and $O_h$, by VECTRI. In each panel, boxed values document the root mean squared difference across land points south of 20°N. In panel (a) coloured rectangles highlight regions of focus including: EA highlands (black); Congolese rainforest (dark green); Guinea coast (dark blue); and South Sudan (purple). Land and country boundaries were added using Natural Earth; free vector and raster map data available at naturalearthdata.com.

simulated precipitation are responsible for increased malaria transmission across parts of the Guinea coast and western regions of central Africa, and decreased transmission risk across the westward side of the EA highlands (Fig 2e). To further understand how differences in temperature and precipitation influence simulated malaria risk, in Fig 3 we show differences in probability distributions on wet days ($\geq 1$ mm day$^{-1}$) in four focus regions (rectangles in Fig 2a). These four regions were chosen as they have different climatological conditions and the effect of explicitly representing convection on simulated malaria transmission varies. Consistent with section S3 in S1 File, there is a greater chance of cooler temperatures and higher precipitation accumulations across all regions in CP4$_h$ (Fig 3). In the Congolese rainforest, Guinea

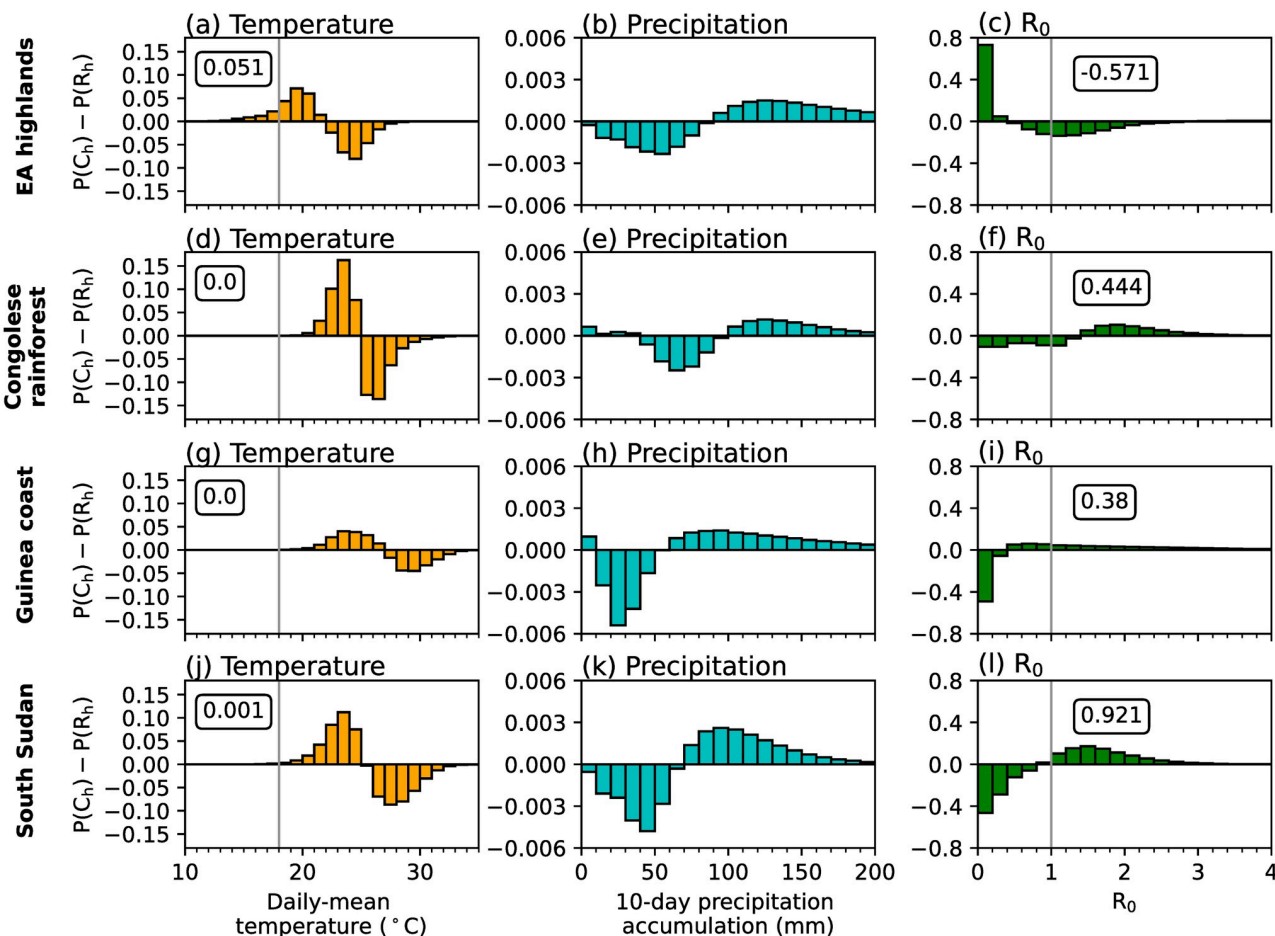

**Fig 3.** Differences in probability distributions of wet-day ($\geq$ 1 mm) grid-point $C_h$ and $R_h$ simulation data in (a,d,g,j) daily-mean temperature (˚C), (b,e, h,k) 10-day precipitation accumulation (mm), and (c,f,i,l) LMM-estimated $R_0$ across (a-c) EA highlands, (d-f) Congolese rainforests, (g-i) Guinea coast and (j-l) South Sudan. Regions are denoted in Fig 2a. In the first column we document the total difference in temperatures changes less than 18˚C, whilst in the third column, we note the total change in days when $R_0$ is greater than 1.0. Both 18˚C and an $R_0$ value of 1.0 are denoted by grey vertical lines.

coast and South Sudan, cooler temperatures (Fig 3d, 3g and 3j) increase the daily survival probability of mosquitoes (not shown) and the number of days when $R_0$ is greater than 1.0 (Fig 3f, 3i and 3l). Across the East Africa highlands on the other hand, cooler temperatures increase the number of days below the sporogonic temperature threshold, which in the LMM is set to 18˚C (S1 Table in S1 File), by 5% (Fig 3a). In the LMM, temperatures below the sporogonic temperature threshold prohibit the development of the malaria parasite and substantially reduces simulated malaria risk. Across East Africa, there is a 57% reduction in the number of days when $R_0$ is above 1.0 (Fig 3c). Although there is substantial differences in simulated precipitation characteristics between $CP4_h$ and $R25_h$ (section S3 in S1 File), we find that differences in simulated malaria transmission when using the LMM are mainly due to temperature differences. Across the three lowland regions, cooler temperatures favour malaria transmission, whilst in highland areas, decreased temperatures reduce the likelihood of surpassing the sporogonic temperature threshold which decreases malaria transmission risk.

Given substantial biases in simulated precipitation and temperature in $CP4_h$ and $R25_h$ (section S3 in S1 File), we also performed LMM experiments which are driven with bias-corrected,

linearly-scaled simulation data (section 2.4). Correcting the mean of simulated rainfall and temperature data substantially reduces differences between LMM experiments (Fig 2i–2k). For example, the RMSD between predicted malaria transmission when driving the LMM with observations compared to $CP4_h$ and $R25_h$ data reduces by 41% and 59% respectively. This indicates that differences between LMM outputs when driving with observations or climate model data are mainly associated with climatological differences rather that variations in atmospheric variability. Additional sensitivity experiments with bias-corrected data show that differences between $C_hBC$ and $R_hBC$ (Fig 2f) are mostly attributable to precipitation differences, except for regions of the Guinea coast where temperature plays a more dominant role (not shown). This suggests that differences in simulated precipitation variability between a parameterised and convection-permitting model lead to larger changes in predicted malaria transmission than differences in temperature variability.

Whilst LMM simulations illustrate that cooler wet-day temperatures in $CP4_h$ increase malaria transmission across lowland regions of Africa, VECTRI outputs show a different behaviour (Fig 2i–2k). Focusing on the difference between $C_h$ and $R_h$, whilst LMM predictions show higher malaria transmission across lowland regions in $C_h$, VECTRI predicts reduced transmission (Fig 2i). We also see a much weaker west-to-east dipole in differences of malaria transmission. We can infer that decreased malaria transmission in $C_h$ relative to $R_h$ is due to a reduced number of simulated wet days (Fig 3b, 3e, 3h and 3k and S2f Fig) and that VECTRI is less sensitive to near-surface air temperatures. The differences between simulated malaria transmission between $C_h$ and $R_h$ when using the LMM and VECTRI highlights that the effect of explicit convection on simulated malaria transmission is sensitive to the chosen mathematical disease model. For the LMM, a substantial sensitivity of malaria transmission to near-surface air temperatures promotes increased malaria transmission across lowland regions, whilst for VECTRI, differences in rainfall frequency lead to decreased transmission. It is noteworthy that VECTRI includes a flushing effect on mosquito breeding sites, and this transmission reduction is consistent with such parameterisation. Consistent with LMM simulations, $C_h$ also has a much stronger agreement with $O_h$ compared to $R_h$. The RMSD between $C_h$ and $O_h$ is approximately 50% smaller compared to the RMSD in $R_h$ (Fig 2i and 2k).

## 3.3 The effect of explicit convection on predicted climate change-induced malaria transmission changes

Given the effect of explicit convection on simulated historical malaria transmission, in this section we investigate whether the representation of convection influences the predicted future change in malaria risk. Fig 4a and 4b show the difference in the annual-mean number of days when $R_0$ is above 1.0 between future and historical LMM simulations. Across the majority of central Africa, parts of the Guinea coast, and regions of south-east Africa, driving the LMM with either climate model configuration predicts reduced malaria transmission under a warmer climate. The largest reductions are observed across the Congolese rainforest with decreases of up to 180 days. Comparing with Fig 1b to 1d, reductions of this magnitude indicate that anthropogenic climate change will decrease the area of Africa where malaria is endemic. However, across the EA highlands, the number of days when $R_0$ is greater than 1.0 is predicted to increase by up to 160 days in both CP4 and R25 configurations. Comparing the difference in simulated malaria transmission changes between the two climate model configurations (Fig 4c) shows that both CP4 and R25 predict a similar decrease in malaria transmission across western lowland regions. Across the EA highlands on the other hand, the magnitude and spatial extent of the increased number of days when $R_0$ is greater than 1.0 is larger in CP4 compared to R25.

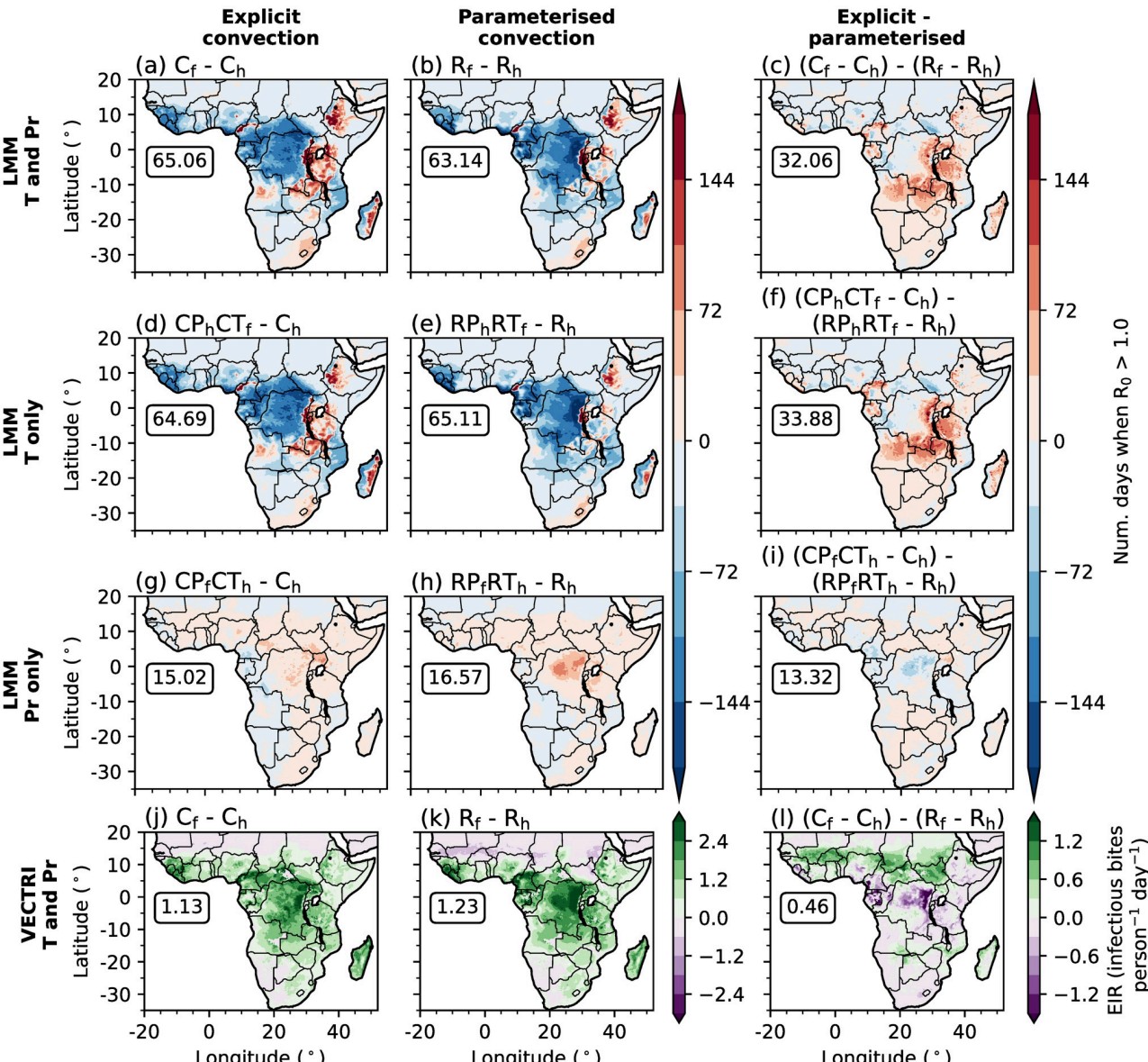

**Fig 4.** (a-i) Differences in the simulated annual-mean number of days when $R_0$ is greater than 1.0 between future and historical LMM experiments. (a-c) Differences when driving the LMM with temperatures and precipitation data from historical and future climates. (d-i) Differences when only using future (d-f) temperature and (g-i) precipitation data. (j-l) Differences in the simulated annual-mean EIR (infectious bites person$^{-1}$ day$^{-1}$) by VECTRI. Differences between future and historical experiments driven by CP4 and R25 data are shown in the first (a,d,g,j) and second (b,e,h,i) columns respectively, whilst panels in the third (c,f,i,l) column show the difference in changes when using CP4 and R25 driving model data. In each panel, boxed values document the root mean squared difference across land points south of 20˚N. Land and country boundaries were added using Natural Earth; free vector and raster map data available at naturalearthdata.com.

To investigate the drivers of simulated changes in malaria transmission under a warmer climate, we perform LMM sensitivity experiments where only temperature (Fig 4d to 4f) or precipitation (Fig 4g to 4i) data is taken from future climate experiments (section 2.4). From these sensitivity experiments we conclude that changes in LMM-simulated malaria transmission are mainly associated with temperature changes for both CP4 and R25 configurations (Fig 4d and 4e). We also find that greater increases in malaria transmission across the EA highlands in

CP4 compared to R25 (Fig 4c) is mostly associated with warmer simulated temperatures (Fig 4f). Precipitation-driven changes in malaria transmission are relatively minimal for both configurations (Fig 4g to 4i) with only a small increase of up to 80 days across the north-east of the Congolese rainforest in R25. Fig 5 shows differences in probability distributions of daily-mean temperatures, 10-day precipitation accumulations and daily LMM-estimated $R_0$ values between historical and future climates in CP4 and R25 across the EA highlands and Congolese rainforest. Increased temperatures across lowland regions in both configurations (Fig 5d and 5j) decrease the probability of mosquito survival. Across the EA highlands on the other hand, anthropogenic climate change increases the probability of daily temperatures exceeding the sporogonic temperature threshold (18˚C; Fig 5a and 5g), which increases the number of days when the malaria parasite is able to develop. As we find that the reduced number of days with a daily-mean temperature below 18˚C across the EA highlands is approximately 5% larger in CP4 compared to R25, we hypothesise that this leads to larger increases in malaria risk under a warmer climate in CP4 compared to R25 (Fig 4c). Although both CP4 and R25 predict higher mean precipitation under a warmer climate (Fig 3), precipitation-driven changes in malaria transmission are relatively small (Fig 4g and 4h).

However, whilst LMM simulations suggest that anthropogenic climate change will reduce malaria transmission across lowland regions of sub-Saharan Africa, VECTRI predicts increased EIRs (Fig 4j and 4k). The larger sensitivity of malaria to precipitation in VECTRI compared to the LMM, leads to larger increases in EIR associated with an intensification of precipitation (Fig 5). Fig 4l shows larger increases in EIR across the Sahel in CP4 compared to R25. This is consistent with findings by [86], who show that only CP4 predicts increased precipitation across the Sahel under anthropogenic climate change. Smaller changes in malaria risk across the Congolese rainforest are associated with a stronger intensification of precipitation in R25 compared to CP4 (Fig 5e and 5k). In summary, the predicted change in malaria transmission under anthropogenic climate change is highly sensitive to the chosen malaria model. For the LMM, for example, the sensitivity to changes in temperature is an artefact of specific model parameterisations. Whilst we find differences in the predicted change in malaria transmission when changing the representation of convection, larger uncertainties are associated with specific malaria model parameterisations.

## 4 Discussion

In this study we investigate to what extent does using atmospheric data from explicit convection simulations affect predictions of malaria transmission risk in historical and future climates. To do so we compare malaria risk experiments driven with climate model data from simulations with either an explicit or parameterised representation of convection. Even though explicitly representing convection improves the frequency and intensity of daily rainfall events [44, 45, 47], the non-linear relationship between local weather conditions and malaria transmission [9, 10, 21] makes it non-trivial to predict changes in malaria transmission when using explicit convection data. This is further complicated by larger errors in near-surface temperatures and 10-day precipitation accumulations when explicitly representing convection (section S3 in S1 File). The concluded impact of explicitly representing convection on simulated malaria transmission is dependent on the chosen malaria model and local climatological conditions. For example, explicitly representing convection increases estimates of transmission risk by the LMM across lowlands in Central Africa, whilst VECTRI simulates a decreased transmission risk. Differences in malaria transmission risk when explicitly representing convection are associated with different parameterisations of thermal and hydrological sensitivities of malaria transmission. The parameterised thermal sensitivity of malaria transmission

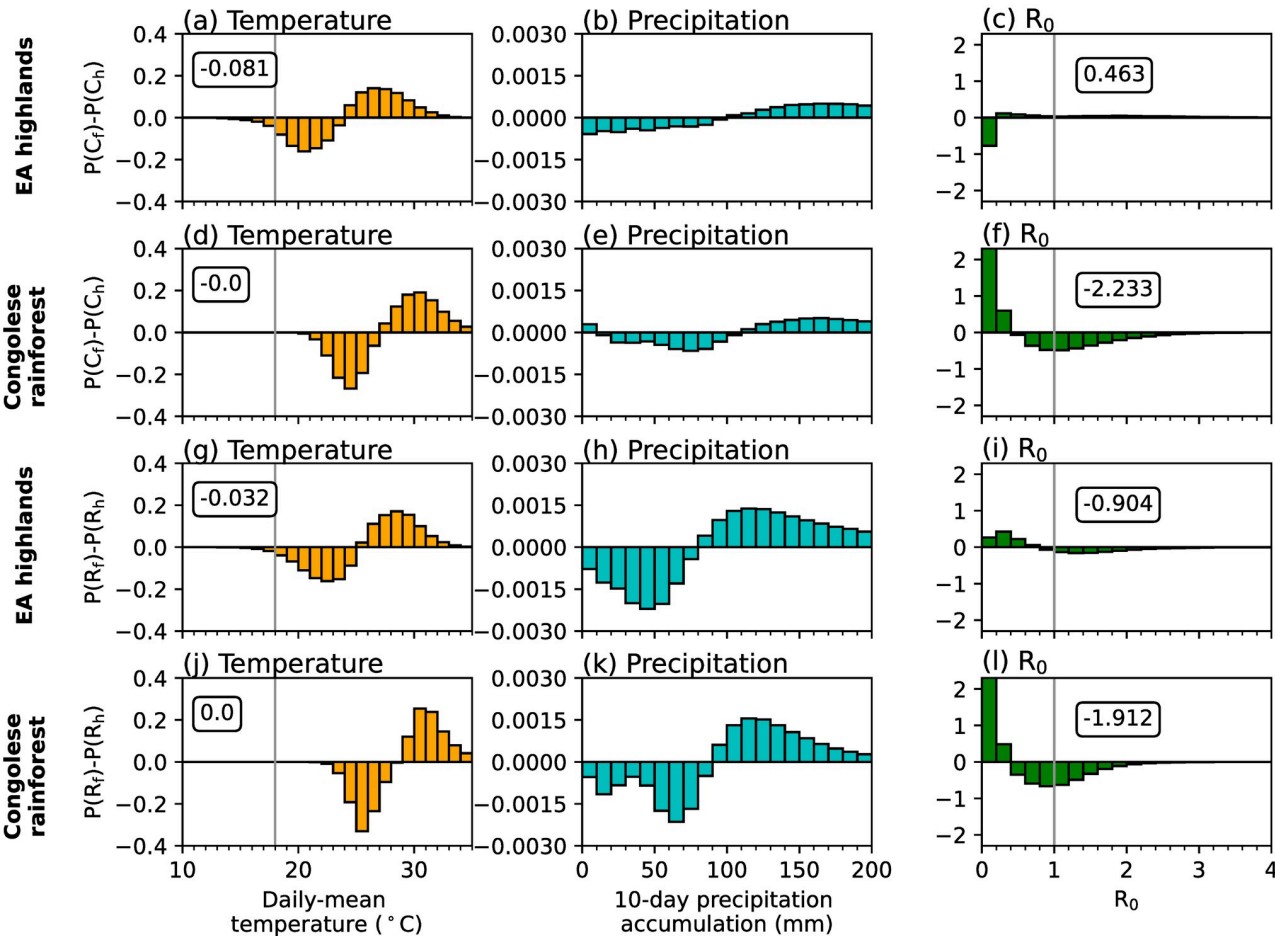

**Fig 5.** Differences in probability distributions of wet-day ($\geq 1$ mm) grid-point future and historical simulation data in (a,d,g,j) daily-mean temperature (°C), (b,e,h,k) 10-day precipitation accumulation (mm), and (c,f,i,l) LMM-estimated $R_0$ across (a-c, g-i) EA highlands and (d-f, j-l) Congolese rainforests. The first (a-f) and last (g-l) two rows show changes in CP4 and R25 experiments respectively. Regions are denoted in Fig 2a. In the first column we document the total difference in temperatures changes less than 18°C, whilst in the last column, we note the total change in days when $R_0$ is greater than 1.0. Both 18°C and an $R_0$ value of 1.0 are denoted by grey vertical lines.

also leads to different behaviours across the East African highlands. In general, explicitly representing convection decreases near-surface temperatures due to higher cloud tops [77]. In the LMM, which is particularly sensitive to near-surface temperatures compared to VECTRI, decreased temperatures reduce the number of days above the sporogonic temperature threshold [8, 15], thereby limiting malaria transmission risk. Differences in the effect of explicitly representing convection on predicted malaria transmission risk reemphasises the uncertainty amongst weather-driven health-impact models [33].

Given the improved representation of tropical rainfall variability when explicitly representing convection (S2 and S3 Figs) [44, 45, 47], we find that malaria models which parameterise a high sensitivity of malaria transmission to rainfall, i.e. VECTRI [31], benefit from improvements in precipitation. For example, the RMSD between predicted malaria transmission when using observations and climate model data is approximately 45% smaller when using data from explicit-convection simulations. The LMM on the other hand, a model where malaria transmission is less sensitive to precipitation [8, 15, 32], only has a 23% improvement. Nevertheless, changes in simulated malaria transmission when using explicit convection model data

for both malaria models highlights an opportunity to use convection-permitting simulations to predict other rainfall-sensitive diseases such as dengue, cholera and typhoid [87–90]. Indeed, future work should assess whether using explicit convection model data changes the conclusion that a warmer climate will increase the transmission of dengue and other arboviruses [91]. Due to significant precipitation errors in parameterised convection simulations [43, 44, 47], caution should be taken when solely relying on low-resolution parameterised-convection climate model data to produce health-relevant adaptation policies.

To explore the role of different atmospheric characteristics in changing simulations of malaria transmission, we performed sensitivity experiments where only temperature or precipitation is changed. Although most studies emphasise improvements in simulated precipitation when explicitly representing convection [47, 77, 92], we find that temperature differences are predominantly responsible for disparities in simulated malaria transmission when using the LMM. Across the majority of sub-Saharan Africa, near-surface temperatures are cooler when explicitly representing convection due to higher cloud tops reducing surface shortwave radiation and near-surface temperatures [77]. This cooling leads to spatial differences in the effect of explicit convection on predicted malaria transmission due to the parameterised relationship between temperature, mosquito survival, and the development of the malaria parasite [8, 15, 16]. Across lowland regions, reduced temperatures increase malaria transmission due to higher mosquito survival rates, whilst in the highlands, decreased temperatures limit the number of days when the sporogonic temperature threshold is surpassed. The sensitivity of malaria transmission to a specified temperature threshold, in our case 18˚C (S1 Table in S1 File), highlights that relatively small temperature perturbations introduced by variations in climate model configuration can lead to substantial differences in climate-sensitive disease modelling. Bias-correcting climate variables removes large differences in estimated malaria transmission risk, indicating that disparities between simulated malaria transmission are predominantly associated with mean climatic conditions. However, we have only tested the sensitivity of simulated malaria transmission to differences in temperature and precipitation using the LMM. We hypothesise that the smaller sensitivity of malaria transmission to temperature in VECTRI compared to the LMM will lead to a smaller temperature effect. Nevertheless, whilst the majority of studies have evaluated precipitation characteristics in convection-permitting simulations [43, 44, 45, 92], due to the use of climate model data in impact-based modelling studies [55–57], future work should also evaluate the representation of other atmospheric variables.

In this study we have shown that differences in climate-dependent malaria model parameterisations result in different effects when using explicit-convection simulation data. In agreement with [33, 37], disparities in climate-dependent malaria model parameterisations lead to differences in the effect of anthropogenic climate change on simulated malaria transmission. VECTRI, a malaria model highly sensitive to precipitation, predicts that the intensification of rainfall driven by anthropogenic climate change [55, 86, 93] will increase malaria transmission across the majority of sub-Saharan Africa, except for the northern fringe of the Sahel where a moderate decreased risk is predicted. The LMM on the other hand, predicts reduced malaria transmission across lowlands, due to higher temperatures and lower probabilities of mosquito survival, and an increase across the East African highlands due to more conducive temperature conditions enhancing the simulated sporogonic cycle. However, we have only investigated the effect of explicit convection on predicted malaria transmission changes using one realisation of a single GCM. Given the substantial uncertainty in predicted rainfall and temperatures changes across Africa in large climate model ensembles [93, 94], and that future atmospheric conditions will depend on sea surface temperature and vegetation changes [95, 96], our conclusions are sensitive to our chosen GCM configuration. Whilst we conclude for some metrics that using climate data from simulations with an explicit representation of convection leads to

a better agreement between malaria simulations driven with observations and climate data, it should not be assumed that using convection-permitting model output will reduce model spread in predicted disease transmission. As climate models evolve to higher-resolutions with different atmospheric parameterisations, efforts should focus on understanding the impact this has on impact-based modelling, reducing the uncertainty amongst impact-based predictions, and assessing the sensitivity and mechanisms of impact disease modelling.

## 5 Conclusions

Malaria is one of the most deadly vector-borne diseases in the world and poses a significant health burden across sub-Saharan Africa. Whilst the disease has been modelled by multiple research groups, our understanding of how different driving climate model data affects simulated malaria transmission remains limited. Over the past decade, increased computational resources have enabled climate simulations to be performed at a resolution which does not require the parameterisation of convective processes. In this study, we use two malaria models, the LMM and VECTRI, to analyse the effect of explicitly representing convection on simulated malaria transmission. Given the improved simulation of certain precipitation characteristics, such as rainfall frequency, when explicitly representing convection, we hypothesised that malaria simulations driven with climate model data from explicit-convection simulations will better agree with observations.

The strength of the conclusion that malaria models driven with climate data from explicit-convection simulations will have a stronger agreement with observations compared to using parameterised convection model data is contingent upon the selected metric and malaria model. As well as this, whilst most studies emphasise that explicitly representing convection improves simulated rainfall characteristics, we find, through sensitivity experiments using the LMM, that simulated malaria differences are mostly associated with temperature variations. We conclude that this behaviour is dependent on the parameterised sensitivity between precipitation and malaria incidence. The different climate-sensitive parameterisations within a malaria model also leads to disparities in the concluded effect of anthropogenic climate change on malaria transmission. When using VECTRI, anthropogenic climate change is projected to increase malaria transmission, regardless of the model configuration with which driving climate data is taken from. Whilst for the LMM, anthropogenic climate change decreases the basic reproductive number across low-land regions. Differences between the LMM and VECTRI highlights uncertainties introduced by different malaria models. Given the increased use of climate model data to produce impact-based forecasts across multiple sectors, including disease transmission, a stringent, detailed and continued assessment of dynamical mechanisms responsible for changes in disease risk is required.

## Supporting information

**S1 File. To support this article we have provided a supporting information document.** The supporting information consists of: detailed descriptions of utilised malaria models (section S1); analysis supporting the choice of observational products (section S2); and simulated atmospheric biases of historical explicit- and parameterised-convection climate integrations (section S3). The document also contains supporting figures; captions of these are below. (PDF)

**S1 Fig. Annual-mean number of days when $R_0$ is greater 1.0 from LMM experiments driven with different observational products.** First, second and third rows are driven with ERA5, CHIRPS and ISIMIP precipitation respectively. Whilst first, second and third columns

are driven with ERA5, BEST and ISIMIP temperature. In all panels boxed values note the spatial correlation coefficient between the annual-mean number of days when $R_0$ is greater 1.0 and MAP data (Fig 1a). To ensure that the spatial correlation is not biased towards regions of low malaria incidence, we remove all grid points where the MAP-derived Pf incidence rate is smaller than 0.1. We also removed grid points where the simulated annual-mean number of days when $R_0$ is greater than 1.0 is outside the range of 15.0 and 140.0. To be consistent with the time span of available MAP data [62], we only compare malaria model output which is driven with climate model data from years 2000 to 2007. All correlations are statistically significant at a 99% confidence interval. Land and country boundaries were added using Natural Earth; free vector and raster map data available at naturalearthdata.com.
(TIF)

**S2 Fig.** Annual-mean differences in (a-c) 10-day precipitation accumulations (mm), (d-f) the number of wet days ($\geq$ 1 mm), (g-i) mean wet-day precipitation rate (mm), (j-l) daily-mean near-surface air temperature (˚C), and (m-o) daily-mean wet-day near-surface air temperature (˚C). Differences are shown between (first column) CP4h and observations, (second column) R25h and observations, and (third column) CP4h and R25h. Values above each panel label, document the root mean squared difference (RMSD) across land points south of 20˚N in each panel. Land and country boundaries were added using Natural Earth; free vector and raster map data available at naturalearthdata.com.
(TIF)

**S3 Fig.** Fractional contributions of (a) daily-accumulated precipitation rates (mm day$^{-1}$) and (b) daily-mean near-surface air temperatures (˚C) across all land points south of 20˚N in bins of 2 mm day$^{-1}$ and 1˚C for (orange) CP4h, (blue) R25h, and (grey) observations. In (a) a subset panel zooms into the fractional contributions of daily-accumulated precipitation rates up to 10 mm day$^{-1}$. A light grey rectangle in panel (a) denotes the area of focus.
(TIF)

# Acknowledgments

Thanks to Professor Christiaan (Tiaan) de Jager (University of Pretoria), Dr Abiodun Adeola (South African Weather Service) and Dr Megan Riddin (University of Pretoria) who co-led the *Investigating the effects of climate change on malaria for urgent action to combat climate change with reference to COP26 priorities* workshop alongside CC, AJ and AM. Without this workshop our collaboration between climate scientists, disease modellers and epidemiologists would not have taken place.

# Author Contributions

**Conceptualization:** Joshua Talib, Remy HoekSpaans, Edmund I. Yamba, Cyril Caminade.

**Data curation:** Joshua Talib, Abayomi A. Abatan.

**Formal analysis:** Joshua Talib, Abayomi A. Abatan, Remy HoekSpaans, Edmund I. Yamba, Temitope S. Egbebiyi.

**Funding acquisition:** Cyril Caminade, Anne Jones, Andrew P. Morse.

**Investigation:** Joshua Talib, Abayomi A. Abatan, Remy HoekSpaans, Edmund I. Yamba, Temitope S. Egbebiyi.

**Methodology:** Joshua Talib, Remy HoekSpaans, Anne Jones, Andrew P. Morse.

**Project administration:** Joshua Talib, Cyril Caminade, Anne Jones, Andrew P. Morse.

**Resources:** Joshua Talib, Abayomi A. Abatan.

**Software:** Joshua Talib, Abayomi A. Abatan, Anne Jones.

**Validation:** Joshua Talib.

**Visualization:** Joshua Talib.

**Writing – original draft:** Joshua Talib, Cyril Caminade.

**Writing – review & editing:** Joshua Talib, Abayomi A. Abatan, Remy HoekSpaans, Edmund I. Yamba, Temitope S. Egbebiyi, Cyril Caminade, Anne Jones, Cathryn E. Birch, Oladapo M. Olagbegi, Andrew P. Morse.

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
