## [Decision Letter · Decision Letter 0]

2 Oct 2023

PONE-D-23-26509The effect of explicit convection on the simulation of malaria transmission across AfricaPLOS ONE

Dear Dr. Talib,

Thank you for submitting your manuscript to PLOS ONE. After careful consideration, we feel that it has merit but does not fully meet PLOS ONE’s publication criteria as it currently stands. Therefore, we invite you to submit a revised version of the manuscript that addresses the points raised during the review process. Although the overall impression of the manuscript was positive and that the quality of the work was sound, there were some concerns that would need to be addressed before being further considered for publication. Specifically, Reviewer 2 raised concerns that the study conclusions were not fully justified by the results. Please ensure that any conclusions presented in the revised submission are fully supported by data from the study.  Please also ensure that the other comments raised by both reviewers are addressed in the revision, and that all data availability adhere's to PLOS One guidelines.

 Please submit your revised manuscript by Nov 16 2023 11:59PM. If you will need more time than this to complete your revisions, please reply to this message or contact the journal office at plosone@plos.org. Please include the following items when submitting your revised manuscript:A rebuttal letter that responds to each point raised by the academic editor and reviewer(s). You should upload this letter as a separate file labeled 'Response to Reviewers'.A marked-up copy of your manuscript that highlights changes made to the original version. You should upload this as a separate file labeled 'Revised Manuscript with Track Changes'.An unmarked version of your revised paper without tracked changes. You should upload this as a separate file labeled 'Manuscript'.

We look forward to receiving your revised manuscript.

Kind regards,

James Colborn

Academic Editor

PLOS ONE

Journal Requirements:

"For this study JT was supported by the Natural Environment Research Council (NERC) as part of the National Capability (NC) international programme (NE/X006247/1). AJ was supported by the Hartree National Centre for Digital Innovation, a collaboration between Science and Technology Facilities Council (STFC) and International Business Machines (IBM)."

"The British Council Researcher Links Climate Challenge Grant (715056901) provided 652

funding for a workshop titled “Investigating the effects of climate change on malaria for 653

urgent action to combat climate change with reference to COP26 priorities” and a 654

follow-on project. Both supported this paper. In particular, thanks to Professor 655

Christiaan (Tiaan) de Jager (University of Pretoria), Dr Abiodun Adeola (South 656

African Weather Service) and Dr Megan Riddin (University of Pretoria) who co-led the 657

aforementioned workshop alongside CC, AJ and AM. Without this workshop our 658

collaboration between climate scientists, disease modellers and epidemiologists would 659

not have taken place. 660

For this study JT was supported by the Natural Environment Research Council 661

(NERC) as part of the National Capability (NC) international programme 662

(NE/X006247/1). AJ was supported by the Hartree National Centre for Digital 663

Innovation, a collaboration between Science and Technology Facilities Council (STFC) 664

and International Business Machines (IBM)."

"For this study JT was supported by the Natural Environment Research Council (NERC) as part of the National Capability (NC) international programme (NE/X006247/1). AJ was supported by the Hartree National Centre for Digital Innovation, a collaboration between Science and Technology Facilities Council (STFC) and International Business Machines (IBM)."

6. We note that [Figures 1, 2, 4, S1 and S2] in your submission contain [map/satellite] images which may be copyrighted. All PLOS content is published under the Creative Commons Attribution License (CC BY 4.0), which means that the manuscript, images, and Supporting Information files will be freely available online, and any third party is permitted to access, download, copy, distribute, and use these materials in any way, even commercially, with proper attribution. For these reasons, we cannot publish previously copyrighted maps or satellite images created using proprietary data, such as Google software (Google Maps, Street View, and Earth). For more information, see our copyright guidelines: http://journals.plos.org/plosone/s/licenses-and-copyright.

a. You may seek permission from the original copyright holder of Figures 1, 2, 4, S1 and S2 to publish the content specifically under the CC BY 4.0 license.  

7. We notice that your supplementary [figures/tables] are included in the manuscript file. Please remove them and upload them with the file type 'Supporting Information'. Please ensure that each Supporting Information file has a legend listed in the manuscript after the references list.

8. Please include a complete caption for your Supporting Information files at the end of your manuscript, and update any in-text citations to match accordingly. Please see our Supporting Information guidelines for more information: http://journals.plos.org/plosone/s/supporting-information. 

Reviewers' comments:

Reviewer's Responses to Questions

**Comments to the Author**

1. Is the manuscript technically sound, and do the data support the conclusions?

Reviewer #1: Yes

Reviewer #2: No

2. Has the statistical analysis been performed appropriately and rigorously? 

Reviewer #1: Yes

Reviewer #2: Yes

3. Have the authors made all data underlying the findings in their manuscript fully available?

Reviewer #1: Yes

Reviewer #2: No

4. Is the manuscript presented in an intelligible fashion and written in standard English?

Reviewer #1: Yes

Reviewer #2: Yes

5. Review Comments to the Author

Reviewer #1: SUMMARY: The authors compare the output of malaria model simulations for sub-saharan Africa when convection is or is not explicitly modeled in a climate dataset. They compare these outputs to those of the MAP malaria data to estimate their accuracy. They then use the two climate datasets for future predictions of malaria to compare how predictions change as a function of accounting for convection. They find that the influence of convection on predictions depends on the model (VECTRI is driven more by changes in extreme precipitation, while LMM is more sensitivity to changes in temperature) and the region of interest. They also find that future malaria predictions are most sensitive to changes in temperature due to convection.

Overall, I think it is interesting study with a well-written manuscript. The majority of my suggestions are minor and focus on improving the structure of how the methods and results are presented. The authors are free to take them into account or not, but I do think clarifying how the different combinations of models and data sources are referred to will increase the manuscript's usability.

STRENGTHS:

The question of which climate data we are using, and how it interacts with the choice of model, is an important question and will be of interest to disease modelers.

The analyses conducted are very comprehensive, containing multiple levels of sensitivity analyses.

The manuscript is well written, and the discussion is very thorough.

MAJOR SUGGESTIONS:

I found the presentation of the methods and results a bit difficult to follow due to the many sensitivity analyses conducted. I make some small suggestions below that could help with this. I also suggest the authors use some form of sub-titles as signposts to help guide the reader to more easily orient themselves to which sensitivity analysis or comparison is currently being addressed. It may also help to divide the figures into ones corresponding to each sensitivity analysis (although I recognize there may be a limit on figures).

line 338: The conclusion that future changes in malaria transmission are primarily due to temperature seems to me that it could also be an artifact of the model used given that LMM is sensitive to changes in temperature input. Some discussion on this should be added and it should be mentioned that this finding is dependent on the model used, and not a generalization that climate change driven changes in rainfall will not impact malaria transmission.

Some further discussion and qualification is needed about using the MAP data as "observed" data, given it is itself the result of a geo-statistical model based on climate variables. Specifically, it may be helpful to mention the variables used in that model (Worldclim for precipitation and MODIS for temperature), how those relate to the two climate models used here, and whether we may a priori expect one to perform better than the other because of this.

MINOR SUGGESTIONS:

Given the wide readership of PLoS One and the audience of this paper, it would be helpful to add a sentence or two further defining convection and the expected impact of it on the study area (like you do in line 420).

line 151: It may be helpful to state explicitly here that CP4 represents a model with convection explicitly modeled and R25 represents the "baseline" model not considering convection (or parameterizing it).

line 192: It may help to make this section clearer by splitting each sensitivity analysis into its own paragraph and adding a topic sentence specifically stating what question that sensitivity analysis is meant to address. I found myself getting a bit lost between the different analyses.

line 196: I found the labeling system a bit hard to follow. One suggestion could be to split Table 2 up by sensitivity analysis or experiment, to make it clearer what combinations of data sources are addressing what scientific questions. It just becomes hard to quickly see what the difference is between two combinations of variables elsewhere in the paper, especially in Figure 2 and 4.

line 201: To reduce confusion and the number of different sensitivity analyses and comparisons, I would move this experiment to the supplement, or make it the main default (See below).

Fig 2: It is a bit confusing having the color bar positive values correspond to blue and negative to red, as the norm is the opposite. Also please state in the figure caption what the numbers in boxes correspond to.

line 284: If the bias-corrected data is a better fit, it seems like that should be used in the primary analysis and the findings comparing non-corrected and corrected in a supplement. Or is the standard in climatology to present the non-bias corrected data? If you choose to keep the non bias-corrected data as the main analysis, I would move the section on bias-corrected to the supplement (see above).

Figure 4: Is there another way you could describe or present panels c,f,i,l? The current labeling system takes some time to interpret and is not easy to understand. Or maybe add a title to this column as you did for the rows?

line 373: These first two sentences could be split up to help with readability. Also the sentence beginning on line 397.

line 535: This link seems to be broken. I think it is missing the tilde (~)

Reviewer #2: The authors examine the difference in the representation of annual mean malaria in Africa in two malaria models; LMM and VECTRI. The models include the influence of meteorology on disease transmission, and the main goal of the study is to assess sensitivity to the meteorological data used to force the model. The data sets tested are output from two regional configurations of the UK Met Office Unified Model (MetUM); one with a convection-permitting 4.5km grid that explicitly represents convection, and the other with a more standard convective parameterization scheme at a horizontal grid spacing of 25km. The main conclusion presented is that the 4.5km model data provides better agreement with the observations and that "future work should utilize high-resolution explicit-convection climate simulations" as a result.

Malaria is one of the most pressing health issues in the world, and understanding how transmission will potentially evolve under climate change is critical. Unfortunately, I have serious reservations about the methodology of the study and its main conclusion. I recommend that the manuscript be returned to the authors for Major and mandatory revisions prior to publication

Major Concerns

As noted above, the manuscript presents as its major conclusion that using a climate model with explicit convection to

drive the malaria model provides superior results to a conventional physics parameterization, and as such "future work should utilize high-resolution explicit-convection climate simulations" (Abstract, last line). However, the evidence presented falls well short of proving or even fully supporting this hypothesis. In Figure 1, the authors do show that there is an improvement in the pattern correlation of annual mean malaria Ro for the LMM for the period 2000-2007 when the convective-permitting configuration (CP4) is used instead of the conventional model (R 25). However, this result a relatively small difference in pattern correlation (0.44 vs 0.37) for a relatively short period, for a single ensemble member of a single climate model for a single malaria model. For the VECTRI model, also presented in Figure 1, the skill is literally identical for CP4 and R25. Given these caveats I do not believe that these results can be generalized to the conclusions the manuscript presents.

The results presented in Figure 2 also call this main conclusion into question. Here the manuscript assesses the impact of temperature and precipitation independently, as well as the effect of climatological biases. Rather than confirming the inherent advantages of the CP4 integrations, Figs. 2g and 2h show that when the climatological biases are removed from each model's output that the R25 actual provides a better comparison to observations. The manuscript notes this result on page 12, stating that "disparities between simulated malaria transmission are predominantly associated with mean climatic conditions". This is a significantly different and less novel conclusion than the claim that explicitly resolving convection leads to a better simulation of malaria transmission. While it is certainly possible, and hoped for, that a more physically accurate representation of convection will lead to a smaller mean bias it is by no means assured.

6. PLOS authors have the option to publish the peer review history of their article (what does this mean?). If published, this will include your full peer review and any attached files.

Reviewer #1: No

Reviewer #2: No

---

## [Author Response · Author response to Decision Letter 0]

14 Dec 2023

Please find response to reviewers attached.

---

## [Decision Letter · Decision Letter 1]

12 Jan 2024

The effect of explicit convection on simulated malaria transmission across Africa

PONE-D-23-26509R1

Dear Dr. Talib,

We’re pleased to inform you that your manuscript has been judged scientifically suitable for publication and will be formally accepted for publication once it meets all outstanding technical requirements.

Kind regards,

James Colborn

Academic Editor

PLOS ONE

Additional Editor Comments (optional):

Reviewers' comments:

Reviewer's Responses to Questions

**Comments to the Author**

1. If the authors have adequately addressed your comments raised in a previous round of review and you feel that this manuscript is now acceptable for publication, you may indicate that here to bypass the “Comments to the Author” section, enter your conflict of interest statement in the “Confidential to Editor” section, and submit your "Accept" recommendation.

Reviewer #1: All comments have been addressed

2. Is the manuscript technically sound, and do the data support the conclusions?

Reviewer #1: Yes

3. Has the statistical analysis been performed appropriately and rigorously? 

Reviewer #1: Yes

4. Have the authors made all data underlying the findings in their manuscript fully available?

Reviewer #1: Yes

5. Is the manuscript presented in an intelligible fashion and written in standard English?

Reviewer #1: Yes

6. Review Comments to the Author

Reviewer #1: (No Response)

7. PLOS authors have the option to publish the peer review history of their article (what does this mean?). If published, this will include your full peer review and any attached files.

Reviewer #1: No

---

## [Editor Report · Acceptance letter]

5 Feb 2024

PONE-D-23-26509R1 

PLOS ONE

Dear Dr. Talib, 

I'm pleased to inform you that your manuscript has been deemed suitable for publication in PLOS ONE. Congratulations! Your manuscript is now being handed over to our production team.

Kind regards, 

on behalf of

Dr. James Colborn 

Academic Editor

PLOS ONE